

# Molecular species composition of polar lipids from two microalgae *Nitzschia palea* and *Scenedesmus costatus* using HPLC-ESI-MS/MS

Nicolas Mazzella[1], Mariem Fadhlaoui[2], Aurélie Moreira[1] and Soizic Morin[1]

[1] INRAE, UR EABX, Cestas, France
[2] INRS-ETE, Québec, Canada

## ABSTRACT

This study examines the polar lipid profiles of two freshwater algae, *Scenedesmus costatus* and *Nitzschia palea*. HILIC-ESI-MS/MS analysis was used to determine and quantify the major phospholipids and glycolipids, as well as their relative molecular species, extracted from the two microalgal cultures. Glycolipids were eluted first, followed by phospholipids partially co-eluting with a sulfoglycolipid. The fragmentation pattern in the negative ionization mode for galactolipids was studied, revealing the stereospecific distribution of fatty acids on the glycerol backbone. Green algae frequently include 18:3 fatty acid in both phospholipids and galactolipids, while monogalactosyldiacylglycerol (MGDG) and digalactosyldiacylglycerol (DGDG) were more saturated and contained shorter acyls. The diatom phospholipids contained mainly molecular species with saturated or monounsaturated fatty acids, while MGDG and DGDG exhibited a higher proportion of polyunsaturated fatty acids, such as the unique and abundant MGDG (20:5/20:2).

## INTRODUCTION

In rivers, periphytic microalgae (*i.e.*, attached to any substrate, either mineral or organic) are often at the bottom of the trophic chain, and plays an important resource for higher trophic level taxa. They typically arise in the form of a biofilm., where organisms from several kingdoms cohabit: besides microalgae, bacteria, fungi and pluricellular micro-eukaryotes are embedded together in a matrix of extracellular polymeric substances. Many herbivores consume biofilms as a primary food source and biofilms, or more specifically their algal component, are receiving increasing attention because they are a source of essential fatty acids for higher consumers (*Taipale et al., 2013*; *Brett & Müller-Navarra, 1997*). They can be used to describe relationships within the aquatic food webs (*Kelly & Scheibling, 2012*) and are often used as indicators of trophic quality in ecosystems. It has long been shown that total fatty acid profiles differ between groups of algae. In freshwater biofilms, which are often dominated by diatoms and chlorophytes, the specific origin influences the overall composition of the food available for consumers, making it a crucial driver of the health

Corresponding author
Nicolas Mazzella,
nicolas.mazzella@inrae.fr,
n.mazzelladibosco@gmail.com

and stability of aquatic food chains. According to *Arts, Ackman & Holub (2001)*, both ω3 eicosapentaenoic acid (EPA, 20:5*n*-3) and docosahexaenoic acid (DHA, 22:6), and the ω6 arachidonic acid (ARA, 20:4*n*-6) are key fatty acids for zooplankton and fish. Freshwater diatoms are also rich in palmitoleic acid (16:1*n*-7), palmitic acid (16:0), and myristic acid (14:0). Most importantly, diatoms are an important source of fatty acids essential for higher consumers, especially EPA and ARA (*Taipale et al., 2013*), and also contain low but significant amounts of DHA (*Demailly et al., 2019*). In contrast, the major fatty acids in chlorophytes are *α*-linolenic acid (ALA, 18:3*n*-3), palmitic acid, oleic acid (18:1*n*-9), and α-linoleic acid (LIN, 18:2*n*-6). Lastly, phosphatidylglycerol and some peculiar galactolipids are the main component of thylakoid membranes in all microalgae (*Mizusawa & Wada, 2012*; *Da Costa et al., 2016*; *Guschina & Harwood, 2006*), and especially their respective fatty acid constituent (*i.e.*, molecular species) were partly investigated for diatom and chlorophyte species so far. Actually, with the exception of a few marine species, lipid algal metabolism has been little studied to date, and we still have limited knowledge of the biochemical processes underlying the synthesis or plasticity of the major lipid classes (*Cutignano et al., 2016*; *Li et al., 2018*).

This is within this framework that we are interested in the analysis of polar lipids of two freshwater microalgae (*Nitzschia palea* and *Scenedesmus costatus*). On the one hand, we aim to characterize the main molecular species associated with both glycolipids and phospholipids, including some fragmentation patterns in mass spectrometry. On the other hand, we aim to quantify each of these compounds. The aim is here to provide a methodology, and then to enlighten the typical polar lipidome profile for both "model" diatom and green algae.

# MATERIALS & METHODS

## Chemicals and materials

The following polar lipid standards were purchased from Avanti Polar Lipids: 1-palmitoyl-2-oleoyl-*sn*-glycero-3-phosphocholine or PC (16:0/18:1) (850457), 1-palmitoyl-2-oleoyl-*sn*- glycero-3-phosphoethanolamine or PE (16:0/18:1) (850757), 1-palmitoyl-2-oleoyl-*sn*-glycero-3-phospho-(1′-rac-glycerol) or PG (16:0/18:1) (840457), 1,2-diheptadecanoyl-*sn*-glycero-3-phosphocholine or PC (17:0/17:0) (850360), 1,2-diheptadecanoyl-*sn*-glycero-3-phosphoethanolamine or PE (17:0/17:0) (830756), 1,2-dipentadecanoyl-*sn*- glycero-3-phosphoethanolamine or PE (15:0/15:0) (850704), and 1,2-diheptadecanoyl-*sn*- glycero-3-phospho-(1′-rac-glycerol) or PG (17:0/17:0) (830456), L-α-phosphatidylserine (Soy, 99%) (sodium salt) (870336) for the phospholipid standards, and monogalactosyldiacylglycerol (840523), digalactosyldiacylglycerol (840524) and sulfoquinovosyldiacylglycerol (840525) from plant extracts as glycolipid standards. Ammonium acetate (LiChropur) were provided by Sigma-Aldrich. Acetonitrile, methanol (MeOH) tert-Butyl methyl ether (MTBE) and isopropanol HPLC grades were purchased from Biosolve Chimie, France. Ultrapure water (UPW) was obtained from Direct-Q® Water Purification System (Merck Millipore, Burlington, MA, USA).

## Algal cultures

The algal cultures were purchased at the Thonon Culture Collection (*Rimet et al., 2018*) under culture references TCC 583 (*i.e.*, the diatom *Nitzschia palea*) and TCC 744 (*i.e.*, the chlorophyte *Scenedesmus costatus*). Both taxa were selected to be common microalgal species in freshwater environments, originated from rivers of the French metropolitan territory. Indeed, the diatom was isolated from the river la Chiers at Longlaville (Eastern France), and the chlorophyte from the stream Foron, a tributary of Lake Léman (France/Switzerland). The algal cultures were incubated in thermoregulated chambers (temperature: 18 °C, light:dark cycle: 16 h:8 h, photosynthetic active radiation reaching the cultures under light conditions: 65 $\mu$mol photons.sec$^{-1}$), in 100-mL Erlenmeyer flask for two exponential growth cycles of seven days before the experiment began. After the two growth cycles aiming at acclimating and synchronizing the cultures, 10 mL of diatom culture (*N. palea*) were put into 60 mL of modified Dauta medium (*Dauta, 1982*) to reach an initial cell density of $297 \pm 75$ cell $\mu$L$^{-1}$. Five replicate cultures were prepared and placed in the thermoregulated chambers under the conditions described above for pre-exposure. In the case of the chlorophyte *S. costatus*, 20 mL of culture and 50 mL of modified Dauta medium (*Dauta, 1982*) were used to obtain an initial cell density of $1,834 \pm 144$ cell $\mu$L$^{-1}$. Cultures were performed in three independent replicates, in the thermoregulated chambers under the same conditions as *N. palea*.

## Lipid extraction

Portions of the following method sections were previously published as part of a preprint (*Mazzella et al., 2023a*). Briefly, 150 mg of microbeads (0.5 mm diameter) were added to 2 mL microtubes along with 10–20 mg (dry mass) of a microalgae culture, which was weighed on a Mettler Toledo NS204S precision balance. Prior to extraction, a 50 $\mu$L solution of PE (15:0/15:0) at 100 ng L$^{-1}$ was added as a surrogate. The extraction method was adapted from *Matyash et al. (2008)* and consisted of adding 1 mL MTBE:MeOH (3:1, v/v) followed by 650 $\mu$L UPW:MeOH (3:1, v/v). A MP Biomedicals FastPrep-24 5G (three cycles of 15 s) allowed homogenization of the solution and mechanical lysis of the sample by the microbeads, thus releasing the analytes from the sample. The upper lipophilic phase (MTBE) was separated from the lower hydrophilic phase (UPW and MeOH) by centrifugation at 12,000 rpm (*i.e.*, 16,000 g). At this point, 600 $\mu$L of the lipophilic phase was collected. After adding 700 $\mu$L of MTBE:MeOH (3:1, v/v) and 455 $\mu$L of UPW:MeOH (3:1, v/v) to the remaining hydrophilic phases, a second extraction (three cycles 15 s) was performed. The supernatant was collected after centrifugation and added to the previous one. Only the collected MTBE phases (about 1.1 mL) were kept for polar lipid analysis. To avoid any enzymatic activity that could contribute to the degradation of the lipid extracts, the entire operation was performed on ice, and butylated hydroxytoluene (BHT, 0.01% (w/v)) was initially added as an antioxidant. A procedural solvent blank was extracted in addition to the samples to confirm that no contamination occurred during the extraction step. All the extracts were stored in a −80 °C freezer. Prior to hydrophilic interaction chromatography coupled to tandem mass spectrometry (HILIC-ESI-MS/MS) analysis, 50 $\mu$L of internal standards (PC, PG, and PE (17:0/17:0)) were added to each sample at a

concentration of 33.3 ng L$^{-1}$. MTBE was evaporated with a stream of N$_2$ and then diluted with an appropriate volume of isopropanol (typically 250 to 1,000 µL).

## HILIC-ESI-MS/MS analysis

Lipid extracts were analyzed with a Dionex Ultimate 3000 HPLC (Thermo Fisher Scientific, Illkirch-Graffenstaden, France). An API 2000 triple quadrupole mass spectrometer (Sciex, Les Ulis, France) was used for detection. Chromatographic separation of both glycolipids and phospholipids was performed on a Luna NH$_2$ HILIC (3 µm, 100× 2 mm) with a Security Guard cartridge NH$_2$ (4× 2.0 mm). The injection volume and temperature column were set to 20 µL and 40 °C, respectively. The chromatographic separation conditions were reported in Table S1, a final pH value of 6.8 was retained for the ammonium acetate buffer and the flow rate was kept constant at 400 µL min$^{-1}$. Further details on initial method optimization and performances related to phospholipid HILIC separation can be found in *Mazzella et al. (2023b)*. Quantitation of phosphatidylcholine (PC), phosphatidylethanolamine (PE) and phosphatidylglycerol (PG) were respectively carried out with PC (16:0/18:1), PE (16:0/18:1), PG (16:0/18:1). Quantitation of glycolipids was carried out with MGDG (16:3_18:3) (63% of the total MGDG standard), DGDG (18:3/18:3) (22% of the total MGDG standard), and SQDG (34:3) (78% of the total SQDG standard). The internal standards utilized were PC (17:0/17:0) for PC phospholipids, PE (17:0/17:0) for PE phospholipids and both MGDG and DGDG glycolipids, and PG (17:0/17:0) for PG phospholipids and SQDG glycolipids. Calibrations curves are provided as (Fig. S2). For glycolipids, a quadratic model was used for the data fitting, while a linear regression was generally used for phospholipids. The limits of quantification for phospholipids and glycolipids are provided in Table S2, they are finally expressed as nmol mg$^{-1}$ (dry weight) since a typical sample of 10 mg of culture is considered for the initial extraction step. Instrumental quantification limits, initially expressed in µg mL$^{-1}$, were determined with signal-to-noise ratios $\geq$ 10. Additionally, PE (15:0/15:0) was used as surrogate for the whole extracting procedure, with a typical recovery of $102 \pm 23\%$ ($n = 10$) (*Mazzella et al., 2023b*). The mass spectrometry was operated at unit resolution and further parameters are reported in Table S3. Multiple-reaction monitoring (MRM) transitions for each molecular species of PC, PG, PE, MGDG, DGDG and SQDG are provided in Table S4. This HILIC-ESI-MS/MS method was adapted from a previous work (*Mazzella et al., 2023b*); however, we have here selected the molecular species representative of microalgae, and thus reduced the number of multiple reaction monitoring (MRM) transitions followed during the two acquisition periods (Table S4). With the exception of SQDG, It was possible to determine the *sn*-1/*sn*-2 ratio for each molecular species of interest by evaluating the relative abundances of the ions corresponding to the two acyl chains.

## Phospholipid and glycolipid nomenclatures

Polar glycerolipids are constituted of a glycerol backbone esterified by two fatty acids on the *sn*-1 and *sn*-2 positions. The moiety linked to the *sn*-3 position refers to the polar head group (*e.g.*, *sn*- phospho-3-glycerol for the PG, a β-D-galactosyl group for MGDG). Each polar head group defines a phospholipid or glycolipid class, and each

class can be divided into several molecular species according to the fatty acyl chain composition and distribution. Polar glycerolipids are abbreviated as follows: when the fatty acyl chain structures are resolved but the *sn*-1 and *sn*-2 positions remain unclear, then the phospholipids or glycolipids are designated PL (C:*n* _C:*n*), with C referring to the sum of the number of carbon atoms and *n* to the number of double bonds for each fatty acyl chain. When the acyl chain composition and distribution are known, then the phospholipids are noted as PL (C:n-1/C:n-2), where C:*n*-1 and C:*n*-2 correspond to the fatty acids linked to *sn*-1 and *sn*-2 positions, respectively (*e.g.*, PG (16:0/18:1) for 1-palmitoyl-2-oleoyl-*sn*-glycero-3-phospho-rac-1-glycerol).

## Conversion of phospholipids and glycolipids in fatty acid equivalents

Following the analysis of the different classes of polar lipids, and having access to the molecular species within each class, it is then possible to deduce the different fatty acids from the acyl chains determined before. To this purpose, each mole of each molecular species was converted into its fatty acid equivalent.

$$1 \; mole \; of \; MGDG \; (18:3/16:3) \rightarrow 1 \; mole \; of \; 18:3 + 1 \; mole \; of \; 16:3 \tag{1}$$

$$1 \; mole \; of \; PG \; (18:1/18:1) \rightarrow 2 \; moles \; of \; 18:1 \tag{2}$$

Equation (1) illustrates the case where the acyl chains are asymmetric (*i.e.*, the fatty acid at *sn*-1 is different from that at *sn-2*), while Eq. (2) corresponds to the other possible case (*i.e.*, the presence of two fatty acids with both the same numbers of carbons and unsaturations).

## RESULTS AND DISCUSSION

Portions of this section were previously published as part of a preprint (*Mazzella et al., 2023a*). From a HILIC-ESI-MS/MS method originally developed for phospholipid analysis (*Mazzella et al., 2023b*), we added the MRM transitions for three classes of glycolipids commonly observed in microalgae (*Li-Beisson et al., 2019*; *Zulu et al., 2018*; *Alonso et al., 1998*): monogalactosyldiacylglycerol (MGDG), digalactosyldiacylglycerol (DGDG) and sulfoquinovosyldiacylglycerol (SQDG). Figure 1 shows the elution orders for the major phospholipids (PC, PE, and PG), as well as the main glycolipids (MGDG, DGDG, and SQDG) obtained from the green algal culture extract. MGDG was eluted first in our conditions, followed by the DGDG around 3.1 min. Finally, it was SQDG that was co-eluted with PG at 6.4 min (Table S2). The absence of any quantifiable amount of phosphatidylinositol (PI) or phosphatidylserine (PS) was uncovered in a first screening of both *N. palea* and *S. costatus* extracts for five phospholipid classes, as well as all possible related molecular species (*Mazzella et al., 2023b*). This result appeared as consistent, since PI, with both PC and PE, is mainly observed in dinoflagellates like *Schizochytrium* sp (*Li et al., 2016*). Afterwards, in addition to the observation obtained with our initial screening (*Mazzella et al., 2023b*) for both *S. costatus* and *N. palea*, the number of MRM transitions followed at the same time for the compounds has been reduced (Table S4). Actually, these transitions have been selected with respect to the molecular species expected

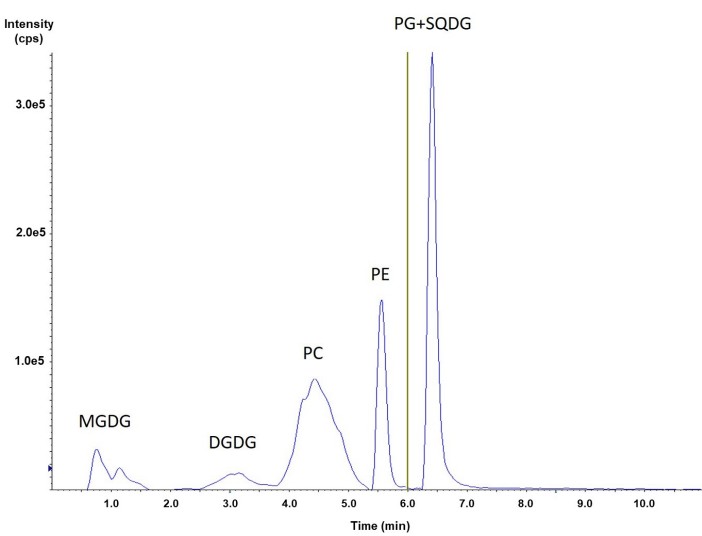

**Figure 1** HILIC-ESI-MS/MS analysis of a phospholipid and glycolipids extracted from *S. costatus*.

for various microalgae (green algae, cyanobacteria, and diatoms), as well as those reported in several previous studies involving lipidomics (*Mazzella et al., 2023b*; *Li-Beisson et al., 2019*; *Zulu et al., 2018*; *Jouhet et al., 2017*; *Coniglio et al., 2021*; *Yongmanitchai & Ward, 1993*; *Degraeve-Guilbault et al., 2017*). With the exception of SQDG, this enabled the consideration of two MRM transitions for each analyte and allowed here the determination of the *sn*-1 and *sn*-2 locations of the acyl chains. However, in order to consider the different MRM transitions associated with each molecular species, it was necessary to determine the likely fragmentation obtained in negative electrospray ionization for such glycolipids.

## MGDG fragmentation pathways

Low-energy collisionally activated dissociation with tandem quadrupole mass spectrometry previously allowed structural characterization of glycerophospholipids, especially in negative ionization mode (*Hsu & Turk, 2009*). Regarding glyceroglycolipids, some studies have focused on the structural elucidation, and thus the determination of fragmentation pathways from positive electrospray ionization mode (*Yingbo et al., 2021*; *Tatituri et al., 2012*). In our case, we focused more precisely on the negative ionization, and the subsequent fragmentation, obtained from a monogalactosyldiacylglycerol standard like MGDG (18:3/16:3) (the fragmentation of DGDG molecular species, not showed here, being similar). The left part of Fig. 2 exhibited the fragments obtained in product scan mode from *m/z* 745.8, which corresponds to MGDG (18:3/16:3), as a deprotonated molecule [M-H]⁻, while the right parts correspond to the precursors determined from the product ions *m/z* 277.5 and 249.3. *Herrero et al. (2007)* have previously observed fragments with *m/z* 277 ratio, and it was attributed to γ-linolenic acid (18:3n- 6) from a MGDG molecular
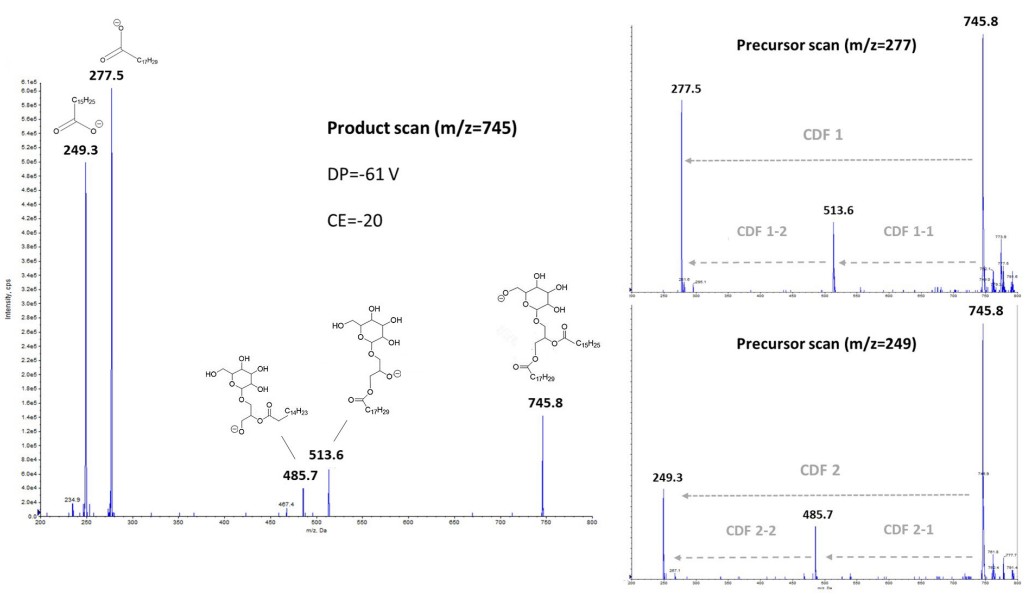

**Figure 2** Negative ionization and collision- induced dissociation of MGDG (18:3/16:3), with either product scanning from *m/z* 745.8 (left part) or precursor scanning for *m/z* 277.5 and *m/z* 249.3 ions (right part).

species in their study. It should be noted that in the case of HPLC-MS/MS analyses, the m/z 277 ratio corresponds more generally to any deprotonated 18:3 isomers.

To explain these observations, it can be assumed that mechanisms such as charge-driven fragmentation (CDF) processes would occur for glycolipids, as for phospholipids during a negative ionization mode (*Mazzella et al., 2005*). Due to the intensity of the two carboxylate ions at *m/z* 249.3 and *m/z* 277.5 formed from the parent ion at *m/z* 745.8 (Fig. 2), a CDF-type mechanism is mainly suspected. In Fig. 3, we proposed two possible pathways with a CDF 1 mechanism inducing the formation of the $R_1COO^-$ ion characterized by a *m/z* 277.5 ratio, and then another CDF 2 mechanism resulting in the formation of the $R_1COO^-$ ion characterized by a *m/z* 249.3 ratio. The dominance of these two CDF mechanisms was confirmed by precursor scanning, which revealed the prevailing m/z 745.8 ratio that also corresponds to the pseudo-molecular ion $[M-H]^-$. In this suggested mechanism, we have assumed an initial localization of the negative charge with the formation of a likely 'alkoxide' ion.

In our case, a second mechanism could also be taken into account with an initial neutral loss of the fatty acyl chain as a ketene (*i.e.*, $R_2CH=C=O$) from *sn*-2 position (CDF 1-1). It can be proposed that the deprotonation at the α carbon on the acyl chain affords an 'enolate' ion (*Fournier et al., 1993*) that immediately undergoes a C-O bond releasing the $C_{14}H_{23}$-CH=C=O neutral, and then affording *m/z* 513.6 (Fig. 4) ion with a negatively charged oxygen atom (*i.e.*, alkoxide ion). A following CDF process (*i.e.*, CDF 1-2) may produce the release of the fatty acyl chain located at *sn*-1 position, with the occurrence of ion at *m/z* 277.5. The alternative pathway consists in the neutral loss of fatty acid a ketene (*i.e.*, $R_1CH=C=O$) from *sn*-1 position (CDF 2-1), and then the final formation

MGDG (18:3/16:3)

m/z=745.9

CDF 1

m/z=249.3

CDF 2

m/z=277.5

**Figure 3** Proposed collision- induced dissociation (CID) pathways of MGDG (18:3/16:3) after electro-spray ionization showing the direct formation of $[R_1COO]^-$ and $[R_2COO]^-$ ions via CDF 1 and 2 processes, respectively.

of the *m/z* 249.3 ion (CDF 2-2). In the Fig. 4, for readability reasons, only a CDF 1-1 pathway, preceding the CDF 1-2 step, with a proton transfer rearrangement was shown. The suggested alternative CDF 2-1 and 2-2 pathways would follow the same principle. Lastly, two other mechanisms might result in the neutral loss of acyl chains as protonated acids (*i.e.*, $R_1COOH$ or $R_2COOH$). In that case, we should observe ions at *m/z* 495.5 and *m/z* 467.5, but these were hardly visible and anyway much less intense than the *m/z* 513.6 and *m/z* 485.7 ions for collision energies tested between 10 and 30 eV.

In this study, whatever the mechanisms and pathways involved, it appears that the most intense product ion corresponds to the fatty acyl chain located at *sn*-1 position, allowing the determination of stereospecific distribution of subsequent fatty acids on the glycerol backbone. Such a result comes into sight as inverted in comparison to the usual negative fragmentation and the latterly observed *sn*-1/*sn*-2 ratio for phosphatidylglycerol, and other phospholipids in general (*Hsu & Turk, 2009*; *Mazzella et al., 2004*; *Hsu & Turk,*

**Figure 4** **Proposed collision- induced dissociation (CID) pathways of MGDG (18:3/16:3) after electrospray ionization showing the likely successive formation of** $[M - -H - -R_2CH = C = O]^-$ **and** $[R_1COO]^-$ **ions (CDF 2-1, and then CDF.**

*2001*; *Pi, Wu & Feng, 2016*; *Huang et al., 2019*; *Hsu & Turk, 2000*). Such a property has been employed here in order to determine the likely regiochemistry of the two acyl chains within the molecular species of MGDG and DGDG, in addition to those associated with PG, PE and PC. This is illustrated by ion ratios observed for the three pairs of molecular species (*i.e.*, possible isomers) of PG (16:0/16:1) and PG (16:1/16:0), PE (16:1/18:1) and PE (18:1/16:1), and DGDG (20:5/16:1) and DGDG (16:1/20:5) (Fig. S1). For instance, the higher intensity of the m/z 253 ratio over the *m/z* 255 ratio, both obtained from the *m/z* 719 ion, indicates the preferential occurrence of the PG (16:0/16:1) regioisomer in that case. Conversely, the higher intensity of the *m/z* 301 ratio over the *m/z* 253 ratio, both originating from the m/z 935 ion, should mainly correspond to the DGDG (20:5/16:1) conformation.

## Intact polar lipids of two microalgae extracts
### Phospholipid and glycolipid molecular species of N. palea and S. costatus

The upper parts of Tables 1 and 2 outline the two microalgae the different molecular species within the detected phospholipids (PE, PG and PC), as well as among the two galactolipids (MGDG and DGDG). In the case of SQDG, fragmentation in negative mode mainly allowed the observation of the characteristic ion at *m/z* 225 (*Hsu & Turk, 2009*; *Mazzella et al., 2007*), and not the couple of ions associated with losses of acyl chains. The expected neutral loss (*Zulfiqar et al., 2003*) (*e.g.*, *m/z* 537.4 with a precursor ion *m/z* 793.5 corresponding to the neutral loss of a 16:0 fatty acid) was barely visible in our case, regardless of the collision energy applied. It was therefore not possible in this case, unlike the other glycerolipids investigated here, to determine the fatty acid composition, resulting

in a general SQDG-type notation ($\sum C : \sum n$), with $\sum C$ the total number of carbon and $\sum n$ the total number of double bonds of the two fatty acids, respectively. Within the phospholipids of *N. palea* (Table 1), it is remarkable that PE and PG contain mainly molecular species with saturated (SFAs) or monounsaturated (MUFAs) fatty acids like 16:0, 16:1, and 18:1. PC differs with many more polyunsaturated fatty acids (PUFAs) such as 20:5 and 22:6. This is also the case for MGDG and DGDG, with the observation of the unique and very abundant MGDG (20:5/20:2) with nearly 2.66 nmol mg$^{-1}$, followed by the four molecular species of DGDG of which three contain 20:5. In accordance with the description in other previous works compounds (*Cutignano et al., 2016*; *Jouhet et al., 2017*; *Yongmanitchai & Ward, 1993*), it should be noted that fatty acids with 16 carbons along with PUFAs, when present, are preferentially positioned in *sn*-2 in this study, reflecting the prokaryotic pathway in the original synthesis of these acylglycerol lipids. Finally, the predominant species of SQDG (32:1) probably corresponds to a combination of 16:0 and 16:1, although this cannot be confirmed analytically due to specific CID fragmentation in our case, as mentioned above.

As reported in Table 2, concerning the green algae (*S. costatus*), we noted, unlike the diatom strain, the frequent presence of 18:3, both among phospholipids and galactolipids, while 20:5 is absent this time. We also find some 16:4 in some molecular species of both MGDG and DGDG. The characteristic MGDG species in most green algae are actually (18:3/16:4), and to a less amount (18:3/16:3), which may be produced by sequential desaturation of MGDG (18:1/16:0) as shown for *Chlamydomonas* or *Dunaliella* (*Giroud, Gerber & Eichenberger, 1988*; *Sanina, Goncharova & Kostetsky, 2004*). DGDG are somewhat more saturated than MGDG, and contains mainly 18:1, 18:2 or 18:3 at *sn*-1 and the shorter 16:0, 16:1, 16:2 and 16:3 moieties at *sn*-2. In addition, the preferential occurrence of 16:3 and 16:4 over 18:3 PUFAs in DGDG compared to PE, PC and PG molecular species was also revealed by lipid content and fatty acid composition of the green alga *Scenedesmus obliquus* (*Choi et al., 1987*) determined with different methodology and analytical techniques (*i.e.*, thin layer and gas liquid chromatography). Consequently, our results appeared to be consistent with these general patterns reported for some *Chlorophyceae*.

Besides, in the case of PG, MGDG and DGDG, when a PUFA is associated with either a MUFA or SFA, it is then mostly found in the *sn*-1 position. The opposite pattern appeared for PE and PC, indicating a likely biosynthesis in the endoplasmic reticulum for these two phospholipids (*Yongmanitchai & Ward, 1993*).

### Polar lipid classes of N. palea and S. costatus

The bottom of Table 1 as well as the bottom of Table 2 summarize the sums of each molecular species, and with these sums, it can be possible to deduce the polar lipid classes for each of the two microalgae. In both cases, MGDG and SQDG are predominant among the glycolipids, followed by PG in the phospholipids from thylakoids. Glycolipids including MGDG, DGDG and SQDG are prevalent in the plant kingdom, representing around 70–85% of membrane lipids in chloroplasts (*Block et al., 1983*). It is well know that thylakoid membrane of chloroplasts is unique in lipid composition, with mono-

**Table 1  Amounts of phospholipid and glycolipid molecular species extracted from *N. palea* (*n* = 5).** Molecular weights, standard deviations (SD) and relative standard deviations (% RSD) are also indicated.

| Molecular species | MW (Da)[a] | Mean (nmol mg$^{-1}$)[b] | SD | % RSD |
|---|---|---|---|---|
| PE(16:0/16:1) | 689 | 0.60 | 0.211 | 35% |
| PE(16:1/16:1) | 687 | 0.76 | 0.350 | 46% |
| PE(18:1/16:1) | 715 | 0.37 | 0.148 | 40% |
| PG(16:1/14:0) | 692 | 0.13 | 0.057 | 46% |
| PG(16:0/16:1) | 720 | 0.56 | 0.213 | 38% |
| PG(16:1/16:1) | 718 | 0.27 | 0.098 | 37% |
| PG(18:1/16:1) | 746 | 0.38 | 0.155 | 41% |
| PG(20:5/18:1) | 794 | 0.10 | 0.000 | 0% |
| PC(20:5/16:1) | 778 | 0.15 | 0.125 | 83% |
| PC(20:5/22:6) | 852 | 0.11 | 0.012 | 11% |
| MGDG(20:5/20:2) | 828 | 2.66 | 1.417 | 53% |
| DGDG(16:1/16:1) | 916 | 0.13 | 0.032 | 24% |
| DGDG(20:5/16:1) | 936 | 0.23 | 0.090 | 40% |
| DGDG(20:5/16:2) | 934 | 0.19 | 0.097 | 50% |
| DGDG(20:5/20:0) | 994 | 0.15 | 0.057 | 38% |
| SQDG(30:1) | 765 | 0.06 | 0.012 | 20% |
| SQDG(32:0) | 795 | 0.08 | 0.014 | 17% |
| SQDG(32:1) | 793 | 1.15 | 0.351 | 31% |
| SQDG(32:2) | 791 | 0.16 | 0.035 | 22% |
| SQDG(34:5) | 813 | 0.06 | 0.010 | 17% |
| SQDG(36:4) | 843 | 0.05 | 0.007 | 14% |
| SQDG(36:5) | 841 | 0.13 | 0.025 | 19% |
| SQDG(42:5) | 925 | 0.05 | 0.008 | 15% |
| ∑ PE |  | 1.73 | 0.692 | 40% |
| ∑ PG |  | 1.43 | 0.497 | 35% |
| ∑ PC |  | 0.26 | 0.137 | 53% |
| ∑ MGDG |  | 2.66 | 1.417 | 53% |
| ∑ DGDG |  | 0.71 | 0.271 | 38% |
| ∑ SQDG |  | 1.74 | 0.439 | 25% |

Notes.
[a] Average molecular weight.
[b] nmol mg$^{-1}$ of dry weight.

and digalactosyldiacylglycerol as major constituents. The ratio of bilayer-forming DGDG to non- bilayer-forming MGDG may affect the properties of chloroplast membranes by altering the lipid bilayer from hexagonal II to lamellar phases (*Demé et al., 2014*). SQDG and PG are classified as acidic lipids, with their negative charge at neutral pH. In addition, acidic lipids with negative charge in their head groups also affect the organization of thylakoid membranes. PG is the only phospholipid produced in the chloroplasts and it is usually an essential component in the center of photosystem II (*Wada & Murata, 2007*). SQDG too has an important function in photosynthesis, although the requirement for this lipid differs among species. PG and SQDG are at least partially functionally redundant, which may be related to maintenance of an anionic charge on the surface of the

**Table 2  Amounts of phospholipid and glycolipid molecular species extracted from *S. costatus* ($n = 3$).** Molecular weights, standard deviations (SD) and relative standard deviations (% RSD) are also indicated.

| Molecular species | MW (Da)[a] | Mean (nmol mg$^{-1}$)[b] | SD | % RSD |
|---|---|---|---|---|
| PE(18:1/16:1) | 715 | 0.03 | 0.01 | 30% |
| PE(18:1/18:1) | 743 | 0.02 | 0.01 | 45% |
| PE(18:1/18:2) | 741 | 0.07 | 0.02 | 30% |
| PE(18:1/18:3) | 739 | 0.17 | 0.05 | 29% |
| PE(18:2/18:3) | 737 | 0.33 | 0.11 | 33% |
| PE(18:3/18:3) | 735 | 0.22 | 0.07 | 31% |
| PG(16:0/16:0) | 722 | 0.03 | 0.00 | 11% |
| PG(16:1/16:0) | 720 | 0.03 | 0.00 | 4% |
| PG(18:1/16:0) | 748 | 0.63 | 0.04 | 6% |
| PG(18:2/16:0) | 746 | 0.37 | 0.03 | 9% |
| PG(18:3/16:0) | 744 | 0.30 | 0.01 | 5% |
| PG(18:4/16:0) | 742 | 0.01 | 0.00 | 4% |
| PG(16:1/18:1) | 746 | 0.12 | 0.01 | 7% |
| PG(16:1/18:2) | 744 | 0.09 | 0.01 | 15% |
| PG(16:1/18:3) | 742 | 0.31 | 0.01 | 5% |
| PG(18:1/18:1) | 774 | 0.01 | 0.00 | 21% |
| PG(18:1/18:2) | 772 | 0.01 | 0.00 | 16% |
| PG(18:3/18:1) | 770 | 0.01 | 0.00 | 9% |
| PG(18:3/18:2) | 768 | 0.01 | 0.00 | 24% |
| PG(18:3/18:3) | 766 | 0.01 | 0.00 | 32% |
| PC(16:0/18:2) | 758 | 0.01 | 0.00 | 9% |
| PC(16:0/18:3) | 756 | 0.07 | 0.02 | 24% |
| PC(18:1/18:3) | 782 | 0.07 | 0.02 | 33% |
| PC(18:2/18:3) | 780 | 0.23 | 0.04 | 18% |
| MGDG(18:3/16:3) | 746 | 0.15 | 0.06 | 41% |
| MGDG(18:3/16:4) | 744 | 3.17 | 0.37 | 12% |
| DGDG(16:0_18:1)[c] | 918 | 0.09 | 0.01 | 11% |
| DGDG(18:2/16:0) | 916 | 0.08 | 0.00 | 1% |
| DGDG(18:3/16:0) | 914 | 0.13 | 0.01 | 6% |
| DGDG(16:1/18:1) | 916 | 0.10 | 0.01 | 15% |
| DGDG(16:1_18:2)[c] | 914 | 0.08 | 0.02 | 22% |
| DGDG(18:3/16:1) | 912 | 0.18 | 0.03 | 18% |
| DGDG(18:1/16:2) | 914 | 0.06 | 0.01 | 8% |
| DGDG(18:2/16:2) | 912 | 0.08 | 0.01 | 13% |
| DGDG(18:3/16:2) | 910 | 0.18 | 0.03 | 19% |
| DGDG(18:1/16:3) | 912 | 0.07 | 0.01 | 18% |
| DGDG(18:2/16:3) | 910 | 0.12 | 0.02 | 19% |
| DGDG(18:3/16:3) | 908 | 0.31 | 0.03 | 9% |
| DGDG(16:4/18:2) | 908 | 0.02 | 0.00 | 18% |
| DGDG(16:4/18:3) | 906 | 0.20 | 0.02 | 12% |

**Table 2** (*continued*)

| Molecular species | MW (Da)[a] | Mean (nmol mg$^{-1}$)[b] | SD | % RSD |
|---|---|---|---|---|
| DGDG(18:3/18:1) | 940 | 0.02 | 0.01 | 70% |
| DGDG(18:3/18:2) | 938 | 0.07 | 0.02 | 32% |
| DGDG(18:3/18:3) | 936 | 0.13 | 0.03 | 22% |
| SQDG(32:0) | 795 | 0.48 | 0.12 | 25% |
| SQDG(32:1) | 793 | 0.07 | 0.01 | 7% |
| SQDG(34:0) | 823 | 0.06 | 0.00 | 9% |
| SQDG(34:1) | 821 | 0.92 | 0.13 | 14% |
| SQDG(34:2) | 819 | 0.49 | 0.08 | 17% |
| SQDG(34:3) | 817 | 1.38 | 0.13 | 9% |
| SQDG(34:4) | 815 | 0.15 | 0.00 | 3% |
| SQDG(36:1) | 849 | 0.05 | 0.00 | 6% |
| SQDG(36:3) | 845 | 0.06 | 0.01 | 13% |
| SQDG(36:4) | 843 | 0.05 | 0.01 | 15% |
| SQDG(36:6) | 839 | 0.13 | 0.01 | 9% |
| $\sum$ PE |  | 0.84 | 0.26 | 31% |
| $\sum$ PG |  | 1.94 | 0.11 | 6% |
| $\sum$ PC |  | 0.38 | 0.07 | 20% |
| $\sum$ MGDG |  | 3.32 | 0.42 | 13% |
| $\sum$ DGDG |  | 1.96 | 0.07 | 3% |
| $\sum$ SQDG |  | 3.84 | 0.49 | 13% |

**Notes.**

[a] Average molecular weight.

[b] nmol mg$^{-1}$ of dry weight.

[c] The intensities of the two product ions associated with the fatty acyl chain were equivalent, and thus the preferential regio-chemistry was undetermined in this case.

thylakoid membrane (*Apostolova et al., 2008*). The amount of phospholipids in microalgae is usually less than that of the glycolipids, with a few exceptions. The main phospholipids are typically phosphatidylcholine, phosphatidylethanolamine, and phosphatidylglycerol. There is a wide variation in the percentages of different found phospholipids in brown algae (*Vyssotski et al., 2017*; *Bergé et al., 1995*), for instance. Furthermore, apart from phosphatidylglycerol, phospholipids are occurring in extraplastidial membranes and among them, phosphatidylcholine and phosphatidylethanolamine are usually the most abundant within algae (*Li-Beisson et al., 2019*), as observed for *N. palea*.

## Polar lipid fatty acid determination from polar lipid molecular species

As illustrated in Eqs. (1) and (2), it is possible to deduce the number of moles of each fatty acid, knowing the number of moles of each molecular species for the combination of the whole polar lipid classes. By summing for example all the 16:1, 18:3 or 20:5 from the various polar lipids they belongs to, a profile of fatty acids from polar lipids (PLFAs) can be estimated for the two microalgae (Table 3).

It was thus obtained two very different profiles, in which we find respectively the predominance of 16:1, 20:2 and 20:5, then to a lesser extent 16:0 and 18:1 for the diatom (*N. palea*), and a majority of 16:4 and 18:3, followed by 16:0, 18:1 and 18:2 for the green algae (*S. costatus*). *Zhang et al. (2020)* showed that six different diatom strains exhibited

**Table 3** Equivalent of polar lipid fatty acids (PLFAs) amounts from the polar lipidome of *N. palea* (left) and *S. costatus* (right), $n = 5$ and $3$ for each microalgae, respectively.

| PLFA | Diatom Mean (nmol mg$^{-1}$)[b] | SD | % RSD | Green algae Mean (nmol mg$^{-1}$)[b] | SD | % RSD |
|---|---|---|---|---|---|---|
| 14:0 | 0.13 | 0.06 | 46% | N.D | N.D | N.D |
| 16:0 | 1.16 | 0.38 | 33% | 1.76 | 0.06 | 4% |
| 16:1 | 4.75 | 1.78 | 38% | 0.94 | 0.09 | 9% |
| 16:2 | 0.19 | 0.10 | 50% | 0.32 | 0.03 | 10% |
| 16:3 | 0.00 | N.D | N.D | 0.65 | 0.06 | 12% |
| 16:4 | 0.00 | N.D | N.D | 3.39 | 0.10 | 12% |
| 18:0 | 0.00 | N.D | N.D | N.D | N.D | N.D |
| 18:1 | 0.85 | 0.30 | 35% | 1.51 | 0.13 | 9% |
| 18:2 | 0.00 | N.D | N.D | 1.56 | 0.16 | 10% |
| 18:3 | 0.00 | N.D | N.D | 6.70 | 0.20 | 5% |
| 18:4 | 0.00 | N.D | N.D | 0.01 | 0.00 | 4% |
| 20:0 | 0.15 | 0.06 | 38% | N.D | N.D | N.D |
| 20:1 | 0.00 | N.D | N.D | N.D | N.D | N.D |
| 20:2 | 2.66 | 1.42 | 53% | N.D | N.D | N.D |
| 20:5 | 3.59 | 1.76 | 49% | N.D | N.D | N.D |
| 22:6 | 0.11 | 0.01 | 11% | N.D | N.D | N.D |
| $\sum$ SFA (%) | 11% | 1% | N.D | 10% | 1% | N.D |
| $\sum$ MUFA(%) | 43% | 2% | N.D | 15% | 1% | N.D |
| $\sum$ PUFA (%) | 46% | 2% | N.D | 75% | 1% | N.D |
| PUFA/(SFA+MUFA) | 0.83 | 0.05 | 6% | 3.01 | 0.32 | 11% |

**Notes.**
[b] nmol mg$^{-1}$ of dry weight.

fatty acid compositions typical of the class *Bacillariophyceae*, with a majority of 14:0, 16:0, 16:1, and 20:5. Among these strains, the authors identified *N. palea* HB170, which showed a fatty acid composition in agreement with our observations, although in our study we focused on polar lipids while *Zhang et al. (2020)* determined the total fatty acid content or that derived from triglycerides alone. Regarding green algae, *Sato et al. (1995)* previously reported the occurrence of pinolenic acid (18:3*n*-6) and coniferonic acid (18:4*n*-3) in the case of *Chlamydomonas reinhardtii*. In our case, we do observe for *S costatus* the presence of 18:3, potentially corresponding to pinolenic acid described earlier, but 18:4, which could be associated with coniferonic acid, was barely detected. Moreover, 18:3 may also correspond to α-linolenic acid (*Kajikawa et al., 2006*). As for 16:4 observed in our green algae extracts, which is typically derived from galactolipids as reported by *Kumari et al. (2013)*, it may correspond to an unusual *n*-3 PUFA also reported with *Chlamydomonas reinhardtii*. The majority of diatom species (or *Bacillariophyta*) investigated in literature (*Alonso et al., 1998*; *Abida et al., 2015*) synthesize glycolipids with a prokaryotic structure, confirming our previous observations. The characteristic fatty acids of diatoms from these glycolipids are 14:0, 16:0, 16:1, and 20:5*n*-3 or eicosapentaenoic acid, whereas 16:2, 16:3, 16:4 18:0, 18:1, 18:2 or 18:3 fatty acids are in general lower or even absent (*Lang et al., 2011*). The desaturation degree and the amount of C16 fatty acids vary in the different diatom species.

Moreover, 16:3 and 16:4 are only present in glycolipids, when occurring, while 20:5 is found in all lipids, but restricted to *sn*-1 position in glycolipids (*Yongmanitchai & Ward, 1993*; *D'Ippolito et al., 2015*; *Xu et al., 2010*), which was consistent with our observations for both MGDG and DGDG (Table 1). To conclude this section, eicosadienoic acid (20:2*n*-6) is a long chain PUFA less frequently observed than eicosapentaenoic in diatoms (*Sayanova et al., 2017*). This fatty acid can be obtained from Δ9 elongation of linoleic acid, and could be considered here as a likely specific marker of *N. palea*.

## CONCLUSIONS

This study analyzed polar lipids of two freshwater microalgae, *Nitzschia palea* and *Scenedesmus costatus*. The main molecular species associated with glycolipids and phospholipids were characterized, including fragmentation patterns in mass spectrometry for glycolipids. Quantification of these molecular species was also provided. The methodology aimed to supply typical polar lipidome profiles for both "model" diatoms and green algae. The most intense product ion corresponded to the fatty acyl chain at the *sn*-1 position, allowing the determination of the stereospecific distribution of subsequent fatty acids on the glycerol backbone for glycolipids, in addition to phospholipids. Polar lipid molecular species of *N. palea* and *S. costatus* also revealed that MGDG and DGDG contained combinations of SFAs or MUFAs with 20:5 for the diatom, while this C20-PUFA was absent from the green algae. The presence of some specific molecular species, such as MGDG (20:5/20:2) in *N. palea*, could be useful in chemotaxonomy to characterize the evolution of algal community structure as a function of environmental stress occurring *in situ* or reproduced with culture conditions. The same polar lipid analysis proposed in this work can of course be further applied to algal monocultures to monitor the physiological effects, on either cytoplasmic membranes or thylakoids, of varying physical (light or temperature), physicochemical (pH, nutrients) or chemical (micropollutants) factors.

## ACKNOWLEDGEMENTS

The authors would like to thank for technical Sylvia Moreira, Gwilherm Jan and Jacky Vedrenne for laboratory assistance regarding microalgae cultures. The authors also thank the Aquatic Vegetation Pole of the ISC XPO (https://doi.org/10.17180/BREY-MR38) of the UR EABX for providing the laboratories and equipment necessary to perform the analyses in this study.

### Funding

This study has been carried out with financial support from the French National Research Agency (ANR) in the frame of the Investments for the future Programme, within the Cluster of Excellence COTE (ANR-10-LABX-45). The funders had no role in study design, data collection and analysis, decision to publish, or preparation of the manuscript.

## Grant Disclosures

The following grant information was disclosed by the authors:
The French National Research Agency (ANR) in the frame of the Investments for the future Programme, within the Cluster of Excellence COTE: ANR-10-LABX-45.

## Competing Interests

The authors declare that there are no competing interests.

## Author Contributions

- Nicolas Mazzella conceived and designed the experiments, performed the experiments, analyzed the data, prepared figures and/or tables, authored or reviewed drafts of the article, and approved the final draft.
- Mariem Fadhlaoui conceived and designed the experiments, performed the experiments, authored or reviewed drafts of the article, and approved the final draft.
- Aurélie Moreira performed the experiments, authored or reviewed drafts of the article, and approved the final draft.
- Soizic Morin conceived and designed the experiments, authored or reviewed drafts of the article, and approved the final draft.

## Data Deposition

The raw data is available Supplemental Files.

## Supplemental Information

Supplemental information for this article can be found online at http://dx.doi.org/10.7717/peerj-achem.27#supplemental-information.

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
