# Peer review of "Molecular species composition of polar lipids from two microalgae *Nitzschia palea* and *Scenedesmus costatus* using HPLC-ESI-MS/MS"

_PeerJ Analytical Chemistry, doi:10.7717/peerj-achem.27_

## Round 0.1 · original submission · Minor Revisions

The reviewers ask for minor revisions. Please try your best to accommodate the suggestions but you don't have to follow if you do not agree. Please provide me and the reviewers the summary of comment-by-comment response that allows one to understand the changes you made without necessarily re-reading the manuscript.

Reviewer 1 ·

Basic reporting

Pass.

Experimental design

Pass.

Validity of the findings

Pass.

Additional comments

Dear Editor,

I have completed a comprehensive review of the manuscript submitted by Mazzella et. al, entitled "Molecular species composition of polar lipids from two microalgae Nitzschia palea and Scenedesmus costatus using HPLC-ESI-MS/MS". This is an intriguing study that provides useful insights in the mass spectrometry-based lipidomics, contributing to our knowledge base in this domain. The authors have clearly put a commendable effort into characterization and quantitation of polar lipids identified in the two freshwater microalgae of interest. The manuscript is largely well-composed, with a clear introduction, structured methodology, and a thorough discussion on results that reflect rigorous scientific integrity. However, there do exist some areas that could benefit from elaboration or refinement.

Major Points/Concerns:
1. The application of the HILIC-ESI-MS/MS method to characterize and quantify major phospholipids and glycolipids is a strong aspect of this work. The technical implementation appears to be sound, and the insights generated are valuable for the field. The authors have demonstrated commendable rigor in their use of internal standards for constructing calibration curves and the application of an internal surrogate to evaluate analyte recovery during extraction, which are both robust methodologies for enhancing the accuracy of LC-MS/MS quantitation. However, the decision to employ an upward curving quadratic model for the calibration curves of SQDG raises concerns (Line 141 and Fig. S2). Typically, in LC-MS/MS analysis, a linear (e.g., PE, PG, and PC calibration curves) or less commonly, a downward curving quadratic model (e.g., MGDG and DGDG calibration curves, representing detector saturation) is expected due to the detector response's relationship with analyte concentration. An upward curving quadratic model is atypical and could suggest potential issues in experimental or data analysis procedures. It is recommended that the authors carefully reassess their SQDG calibration data and revise the regression model, thereby improving the reliability and accuracy of their SQDG quantitation results.

2. The authors' exploration of the fragmentation pathways of MGDG using tandem quadrupole mass spectrometry, particularly in negative ionization mode, represents a significant contribution to lipidomics studies. The detailed elucidation of fragmentation pathways, with a specific focus on the negative ionization and subsequent fragmentation of an MGDG standard, is highly commendable. The authors' approach, proposing robust charge-driven fragmentation (CDF) processes and interpreting the formation of specific ions, offers valuable insights. The thorough investigation into observed and potential pathways, with keen attention to the intensity of specific ions, reflects the strength of their methodology. The determination of the stereospecific distribution of subsequent fatty acids on the glycerol backbone provides crucial understanding of the structure and fragmentation behavior of glyceroglycolipids. Moreover, the authors bring to light an intriguing inversion in fragmentation when compared to commonly studied phospholipids. The comprehensive investigation into the structural elucidation of MGDG and the discernment of the regiochemistry of acyl chains within molecular species of MGDG, DGDG, PG, PE, and PC, represents a noteworthy advancement in the field.
3. The observations on the lipid content of the diatom are interesting, notably the prevalence of saturated or monounsaturated fatty acids in phospholipids compared with a higher degree of polyunsaturated fatty acids in MGDG and DGDG. This contrast invites further investigation and discussion.

Minor Points/Concerns:
1. It appears that a dedicated conclusion section is absent from this manuscript. Such absence may prevent a clear summarization of the principal findings, their significance, and the implications for future research. I would recommend the authors to add a conclusion section to enhance the overall structure and coherence of the manuscript and facilitate readers in understanding the broader impact of the study.
2. While the context of this study is well presented, providing more background on the potential implications of these findings could widen the appeal of this work. Consider explaining how these specific lipid profiles could inform broader biological studies or have potential applications in industrial settings.
3. The authors are encouraged to offer more in-depth interpretation of their results. The potential biological roles and implications of the identified lipid species, the reasons behind the higher saturation of DGDG compared to MGDG in the green alga, and the potential impact of these findings on the microalgae's functionality warrant a more detailed exploration.
4. Line 92: “erlenmeyers" can be rephrased as ““Erlenmeyer flasks"
5. Line 131 and Table S1: The HPLC flowrate is not mentioned. The authors may consider reporting the HPLC flowrate.
6. Line 147: “Q1 > Q3 transitions” can be rephrased as “MRM transitions”. Also define the acronym “Multiple-reaction monitoring (MRM)” to help non-mass spectrometrist readers better understand the term.
7. Line 187 and 188: the term “, it was proceed…” seems to be a typo.
8. Line 203 and Figure 2: To better demonstrate the sn-1 and sn-2 acyl chains with their corresponding product ions, it might be helpful to put precursor and product ion structures next to corresponding m/z peaks on the mass spectra.
9. Line 233 and Figure 3: the initial localization of the negative charge on MGDG (18:3/16:3) precursor ion may not be the same as depicted on Figure 3. The authors may want to include additional literature support for the proposed initial structure, or mention that it is a putative structure, next to the introduction of “pseudo-molecular ion [M-H]-.”
10. Table 1 and 2: I would assume that “SD” stands for “standard deviation”. Such acronym definition is appreciated here.
11. Figure S1: “…couple of analytes”
12. Table S2 and Line 142: There are different ways to calculate the LoQ, but a commonly used method is based on the standard deviation of the response (σ) and the slope of the calibration curve (S). It would be helpful if the authors could elaborate a little more on this topic.
13. Table S3: add a unit for the “collisionally activated dissociation” gas pressure. Is it in psi?
14. Table S4: add units for all the column titles: Q1 (m/z), Q3 (m/z), Dwell time (ms), DP (V), and CE (V).
15. Supplementary Info: add page numbers.

In conclusion, while the manuscript makes important contributions to the field of microalgal lipidomics, it could be enhanced through minor revisions before publication on PeerJ.

Best regards,
Reviewer

Reviewer 2 ·

Basic reporting

This manuscript by Mazzella et al. presents the profiles of polar lipids for two freshwater algae, the diatom and a common green alga, using the HPLC-ESI-MS/MS analysis. The HPLC elution profiles and the ESI mass spectra and fragmentation patterns of lipids from the algae extracts provide rich information regarding the classes, molecular species as well as regiochemistry of the lipids, specifically phospholipids and glycolipids. It would interest readers in the microalgae, lipids, and mass spectrometry communities. It is also at the right level for a PeerJ Chemistry article. I have listed a few questions below as I read through the manuscript. I encourage the authors to consider these queries before the manuscript is published.

General questions:
a) I have highlighted sentences difficult to read, please see them in my attached document. I suggest the authors ask for proofreading from native English speakers or professional English editors.
b) Some abbreviations familiar to the mass spectrometry community may confuse and keep away some readers outside the area. I suggest the authors give a short annotation to the following abbreviations upon their first appearances: HILIC, expression of fatty acids like 20: 5n-3 and (16:0/18:1), Q1 > Q3 transitions, and MRM transitions.
c) In Figure 1 and 2, the labels of axis ticks are difficult to read. I suggest the authors enlarge them.
d) I think a conclusion section is a requirement for the PeerJ Chemistry journal, but I did not find the conclusion part in the manuscript. I suggest the authors add one conclusion section and summarize the findings as well as their significances.

Questions for the Abstract:
Line 14: Only after searching online, I realized HILIC is one category of HPLC. I know HPLC well but have no idea about HILIC. So, I think it might not be appropriate to put HILIC in an abstract. Instead, HPLC could be a better option. As I said above, the authors could give a short annotation on HILIC in the main text.

Questions for the Introduction:
Line 39: “food webs” could be a more appropriate term to use than “foodweb”

Line 40: Could the authors provide more details about “total fatty acid
profiles”. What do the profiles include?

Line 42: It is unclear to me what the distribution refers to specifically.

Questions for the Materials & Methods section:
a) I suggest the authors add a figure to show the experimental scheme for HILIP-ESI-MS/MS, and declare more clearly which quantity can be obtained from each step. For example, in Line 136, the authors claimed quantitation of glycolipids, but what is quantified specifically?
b) What is the mass resolution of the mass spectrometer?
c) I suggest the authors add a sentence or two to explain why PC/PE/PG and MGDG/DGDG/SQDG were chose for the quantitation.
d) As I mentioned above, it would be better to have short annotations for Q1 > Q3 transitions and MRM transitions.
e) Line 151: excepted should be a typo.
f) Line 151 – 153: It is not clear to me how the sn1/sn2 ratio can be derived from the relative abundances of ions respective to the two acyl chains. Also, the word “searched” at the end of line 151 seems unreasonable.
g) Line 161: could the authors be more specific about fatty acyl chain distribution here?

Questions for the Results and discussion section:
Figure 1:
a) The vertical yellowish green line confuses me a lot. I realized after I read the main text that PG and SQDG were co-eluted. I was thinking the PG peak was the tiny one located around the vertical line.
b) I am curious how the authors quantified PG and SQDG since they were inseparable in HPLC.
c) I wonder why the authors only put the elution profile of the green algae extract. Could the authors also include the elution profile of the diatom algae extract in Figure 1?

Line 199: at the end of this line, the word “reduced” does not make sense to me. Do the authors mean to say “deduced”?

Line 231-233: the last sentence of this paragraph is difficult to read due to some grammar issues. I suggest the authors re-write it.

Experimental design

no comment

Validity of the findings

no comment

Annotated reviews are not available for download in order to protect the identity of reviewers who chose to remain anonymous.

Reviewer 3 ·

Basic reporting

It's ok

Experimental design

It's ok

Validity of the findings

It's ok

Additional comments

In this manuscript, the authors describe the polar lipid composition of two microalgae Nitzschia palea and Scenedesmus costatus by HPLC-ESI-MS/MS. The introduction is well written and the methodology is clear, allowing other researchers to test the methods used to assess the quality of extracts from other species. The results are interesting, original and valuable and my opinion is favorable for publication in your journal after minor revision.
comments:
- The abstract is too long, rephrase it, and add keywords.
- Add graphic abstract
- Figure 1: change the title, it is not an ESI-MS/MS spectrum, it is a chromatogram.
-Figures 3 and 4: The chemical structures must be drawn uniformly (nucleus...)
- A complete review of the English writing style is also recommended.
-The references will have to be rechecked: all the newspapers must be in abbreviation.

---

## Round 0.2 · Minor Revisions

All reviewers are satisfied with the revision, but there is some overlapping text in the methods section which needs to be addressed in the final version.

Reviewer 1 ·

Basic reporting

It's okay. No more comments.

Experimental design

It's okay. No more comments.

Validity of the findings

It's okay. No more comments.

Reviewer 2 ·

Basic reporting

pass.

Experimental design

pass.

Validity of the findings

pass.

Additional comments

The authors have addressed my questions/comments excellently. I believe it is appropriate for the PeerJ Analytical Chemistry.

---

## Round 0.3 · accepted · Accept

Congrats! It seems good to publish in the current version.